# Prognostic Significance of Pseudotime from Texture Parameters of FDG PET/CT in Locally Advanced Non-Small-Cell Lung Cancer with Tri-Modality Therapy

**DOI:** 10.3390/cancers14153809

**Published:** 2022-08-05

**Authors:** Hyunjong Lee, Hojoong Kim, Yong Soo Choi, Hong Ryul Pyo, Myung-Ju Ahn, Joon Young Choi

**Affiliations:** 1Department of Nuclear Medicine, Samsung Medical Center, Sungkyunkwan University School of Medicine, Seoul 06351, Korea; nmhjlee@gmail.com; 2Division of Pulmonary and Critical Care Medicine, Department of Medicine, Samsung Medical Center, Sungkyunkwan University School of Medicine, Seoul 06351, Korea; hj3425.kim@samsung.com; 3Department of Thoracic and Cardiovascular Surgery, Samsung Medical Center, Sungkyunkwan University School of Medicine, Seoul 06351, Korea; ysooyah.choi@samsung.com; 4Department of Radiation Oncology, Samsung Medical Center, Sungkyunkwan University School of Medicine, Seoul 06351, Korea; hr.phy@samsung.com; 5Division of Hematology-Oncology, Department of Medicine, Samsung Medical Center, Sungkyunkwan University School of Medicine, Seoul 06351, Korea; silk.ahn@samsung.com

**Keywords:** non-small cell lung cancer, FDG PET/CT, texture analysis, prognosis, pseudotime analysis

## Abstract

**Simple Summary:**

Although texture parameters of F-18 fluorodeoxyglucose positron emission tomography/computed tomography images were known to associate tumor biology and clinical features, the types and implications of parameters are too various and complicated. To overcome the limitation of texture parameter, we attempted to produce a new simplified parameter from texture parameters of F-18 fluorodeoxyglucose positron emission tomography/computed tomography images in lung cancer patients using pseudotime analysis. Pseudotime analysis is a recently developed method to explore changes in cell or tissue characteristics based on transcriptomic expression. It is the first study to apply pseudotime analysis into radiomics dataset other than transcriptomics data. Herein, we demonstrated that pseudotime can be successfully estimated from texture parameters. In the aspect of prognostic prediction, pseudotime was an independent prognostic factor for overall survival in contrast to conventional parameters such as metabolic tumor volume and total lesion glycolysis. This study showed possibility of integrating various texture parameters into single parameter which reflects disease progression status. Pseudotime, as a concrete value of disease progression, is expected to be used in clinical field to evaluate disease and predict prognosis.

**Abstract:**

Texture analysis provides image parameters from F-18 fluorodeoxyglucose positron emission tomography/computed tomography (FDG PET/CT). Although some parameters are associated with tumor biology and clinical features, the types and implications of these parameters are complicated. We applied pseudotime analysis, which has recently been used to estimate changes in individual sample characteristics, to texture parameters from FDG PET/CT images of locally advanced non-small-cell lung cancer (NSCLC) patients undergoing neoadjuvant concurrent chemoradiation therapy (CCRT) followed by surgery. Our subjects were 303 NSCLC patients who underwent pretherapeutic FDG PET/CT and tri-modality therapy. Texture parameters of the primary tumor were calculated from FDG PET/CT images acquired before neoadjuvant CCRT. Pseudotime analysis was performed using the PhenoPath tool. Clinicopathologic features including survival data were collected and survival analysis was performed to compare the prognostic significances of pseudotime parameters with those of conventional PET parameters. Pseudotime was successfully estimated from texture parameters. Normalized co-occurrence homogeneity, normalized co-occurrence inverse difference moment, and black–white symmetry showed positive correlations with pseudotime, short run emphasis, normalized co-occurrence dissimilarity, and short zone emphasis negative correlation. The maximum standardized uptake value (SUV) and mean SUV were not associated with overall survival. Pseudotime, metabolic tumor volume (MTV), and total lesion glycolysis (TLG) showed significant associations with overall survival. In contrast to MTV and TLG, pseudotime was an independent prognostic factor for overall survival. Various metabolic texture parameters can be integrated into a single parameter using pseudotime analysis. Pseudotime of the primary tumor, estimated from FDG PET/CT images, better predicts overall survival in locally advanced NSCLC patients treated with tri-modality therapy than conventional PET parameters.

## 1. Introduction

F-18 fluorodeoxyglucose positron emission tomography/computed tomography (FDG PET/CT) is a robust imaging modality to diagnose and determine the appropriate management of lung cancer [1]. The most representative parameter, maximum standardized uptake value (SUVmax), provides diagnostic thresholds to identify malignant nodules in the lung [2] and is an excellent prognostic factor for both disease-free survival (DFS) and overall survival (OS) in lung cancer [3]. FDG PET/CT has special clinical significance in locally advanced lung cancer, which can be treated with various therapy options. Beyond conventional parameters such as SUVmax, volumetric parameters such as metabolic tumor volume (MTV) and total lesion glycolysis (TLG) have demonstrated good prognostic significance in locally advanced non-small-cell lung cancer (NSCLC) [4,5].

Radiomics, which produces texture features from medical images using diverse algorithms, is a recently highlighted concept in nuclear medicine [6]. Beyond conventional metabolic and volumetric parameters, texture parameters are associated with tumor biology and prognosis in lung cancer patients. For example, entropy and homogeneity are significant prognostic factors for progression-free survival of lung cancer patients [7]. However, radiomics has limitations of diversity and complexity due to the variety and complicated nature of the analysis algorithms. Among varying parameters, it is difficult to extract significant texture parameters associated with cancer diagnosis or prognosis with high reproducibility. In addition, it is difficult to understand the implications or meanings of each texture parameter. A single simplified parameter integrating various texture parameters would be better for applying radiomics in the clinical field.

Pseudotime analysis, also called trajectory inference analysis, is a spotlighted method to explore changes in cell or tissue characteristics based on transcriptomic expression [8]. It provides a numerical scale to reflect where a cell or tissue is in the course of disease. A previous study investigated cellular dynamics in lung cancer using pseudotime analysis [9]. In another, a tissue-scale RNA-sequencing dataset was successfully analyzed by pseudotime analysis to demonstrate the evolution of tumor characteristics in lung cancer [10]. However, no previous study has applied pseudotime analysis to radiomics data. We hypothesize that pseudotime analysis can be applied to radiomics datasets to estimate relationships or the temporal evolution of medical images from cancer patients. As RNA-sequencing data corresponds with radiomics data and each cell or tissue sample corresponds with an individual patient, pseudotime analysis can be implemented into texture parameters.

In this study, we acquired metabolic texture parameters from FDG PET/CT of NSCLC patients. Pseudotime analysis was conducted for the texture parameters of the primary tumor. The prognostic value of our newly developed pseudotime parameter was compared with that of conventional metabolic and volumetric parameters of FDG PET/CT in patients with locally advanced NSCLC undergoing tri-modality therapy.

## 2. Methods

### 2.1. Subjects

Four hundred fifty-nine consecutive patients undergoing FDG PET/CT examination for the initial staging of NSCLC and subsequent neoadjuvant concurrent chemoradiation therapy (CCRT), between January 2008 and December 2020, were retrospectively enrolled. Among them, 24 patients with pathologies other than adenocarcinoma or squamous cell carcinoma and 27 patients who did not undergo curative surgery after neoadjuvant CCRT were excluded. One patient with clinical stage IV disease and 17 patients with clinical stage N1 or N3 disease were excluded to limit the sample to patients with the same clinical stage. Five patients who did not undertake FDG PET/CT after neoadjuvant CCRT were also excluded. Based on previous studies suggesting that the minimum MTV eligible for radiomics analysis in FDG PET/CT is approximately 10 cm^3^, 82 patients with metabolic tumor volumes smaller than 10 cm^3^ were excluded [11,12]. Finally, 303 patients who underwent tri-modality therapy were included in this study (Figure 1). Our institutional review board approved this retrospective cohort study (#2022-01-086), and the requirement for informed consent was waived.

### 2.2. FDG PET/CT Acquisition

All patients fasted for at least six hours and had blood glucose levels of less than 200 mg/dL at the time of their FDG PET/CT scans. Whole-body PET and CT images from basal skull to mid-thigh were acquired 60 min after the injection of 5.0 MBq/kg FDG without intravenous or oral contrast on a Discovery LS, a Discovery STE, or a Discovery MI DR PET/CT scanner (GE Healthcare, Milwaukee, WI, USA). Continuous spiral CT was performed with an 8-slice helical CT (140 keV, 40–120 mA; Discovery LS) or 16-slice helical CT (140 keV, 30–170 mA; Discovery STE, 120 keV, 30–100 mA; Discovery MI DR). An emission scan was then obtained from head to thigh for 4 min per frame in the 2-dimensional mode (Discovery LS), 2.5 min per frame in the 3-dimensional mode (Discovery STE), or 2 min per frame in the 3-dimensional mode (Discovery MI DR). PET images were reconstructed using CT for attenuation correction by the ordered-subsets expectation maximization algorithm with 28 subsets and 2 iterations (matrix 128 × 128, voxel size 4.3 × 4.3 × 3.9 mm; Discovery LS), ordered-subsets expectation maximization algorithm with 20 subsets and 2 iterations (matrix 128 × 128, voxel size 3.9 × 3.9 × 3.3 mm; Discovery STE), or ordered-subsets expectation maximization algorithm with 18 subsets and 4 iterations (matrix 192 × 192, voxel size 3.9 × 3.9 × 3.3 mm; Discovery MI DR). SUV was calculated by adjusting for administered FDG dose and the patient’s body weight.

### 2.3. FDG PET/CT Image Analysis

Image feature extraction was based on the threshold segmentation method with a threshold SUV value of 2.5 in MIM version 6.4 software (MIM Software Inc., Cleveland, OH, USA). Briefly, the target primary tumor was identified by an experienced nuclear medicine physician who was unaware of all of the clinical information, except the target tumor site. As the physician dragged the cursor out from the center of the target tumor to a point near the edge of the lesion, the software automatically outlined a three-dimensional volume of interest above the SUV of 2.5 on the tumor. After creating segmentation of the target tumor lesion, we extracted PET image features using the Chang Gung image texture analysis toolbox (CGITA, https://code.google.com/p/cigita, accessed on accessed on 1 July 2021), an open-source software package implemented in MATLAB (version 2012a; MathWorks Inc., Natick, MA, USA) [13]. A total of 86 PET features available in CGITA were measured on each segment. Parameters from the voxel alignment matrix, neighborhood intensity difference, intensity size zone matrix, normalized co-occurrence matrix, and neighboring gray level dependence, and SUV statistics, except conventional parameters, were selected as input data. Conventional parameters such as SUVmax, mean SUV (SUVmean), MTV, and TLG were also calculated by the CGITA software.

### 2.4. Pseudotime Estimation

A pseudotime trajectory was generated using the “PhenoPath” package in R [14]. PhenoPath, an analytical tool for pseudotime, has previously been used to estimate the ordering of gene expression measurements across individual objects. It employs Bayesian statistics and models latent progression of gene expression. In this study, PhenoPath is applied to the radiomics dataset, which corresponds to the RNA-sequencing data described in previous reports. For preprocessing, ComBat harmonization was conducted to remove batch effects due to various PET/CT instruments using the package “neuroCombat” in R. Subsequently, corrected texture parameters were normalized using the “scale” function in R. The input data were a normalized texture parameter matrix of initial FDG PET/CT images from 303 lung cancer patients. We chose an evidence lower bound (ELBO) of 10^−6^ and computed thinned by 2 iterations. Then, the PhenoPath algorithm repeated the calculation to predict pseudotime with 2 iterations until ELBO reached below 10^−6^. ELBO is a quantity to reflect optimized approximation in probabilistic inference [15]. Ultimately, pseudotime was estimated as a reference value for latent progression of the texture characteristics from FDG PET/CT. Pseudotime was normalized with range of 0 to 1 for further analysis.

### 2.5. Clinical Variables and Follow-Up

Clinical information including sex, age, performance of adjuvant therapy, and histological type of the primary tumor was obtained by reviewing electronic medical records. Radiologic reports of CT covering the chest were reviewed, and the locations of the primary tumors were obtained. Clinical tumor stage (cT stage) was evaluated by the size of the primary tumor measured by CT of the FDG PET/CT scans. Clinical nodal stage (cN stage) was evaluated by the maximal number of lymph nodes positive for metastasis on the CT or FDG PET/CT scans. After we reviewed the pathologic reports of surgical specimens, pathological T and N stages were determined based on the AJCC/UICC staging system (8th edition).

Adjuvant therapy was performed after surgery according to each patient’s situation and their corresponding physician’s decision. After surgery, all patients were monitored regularly to obtain accurate information regarding recurrence. The follow-up program was every 2–4 months during the first year, every 4–6 months during the next 2 years, and every year thereafter. Every follow-up evaluation included a complete physical examination, complete blood count, biochemical screening, and chest X-ray. CT scans of the chest were performed from every 6 months to 1 year, or more frequently if clinically indicated.

Recurrence or metastasis was considered when there was an abnormal finding suggesting recurrence or metastasis on serial imaging studies or pathologically confirmed malignancy. The events for survival analysis were defined as recurrence or metastasis and any cause of death. The disease-free and overall survival durations from the last follow-up or event were recorded for each patient.

### 2.6. Statistical Analysis

Correlation analyses were performed to reveal associations between image parameters and estimated pseudotime. Pearson’s correlation analysis was performed for each image parameter and pseudotime. Age was recorded as a continuous scale and divided into three groups as a discrete scale according to tertiles for log-rank tests and multivariate analyses. Clinical variables including sex, age by both discrete and continuous scales, location of primary tumor, cT stage, clinical TNM stage, performance of adjuvant therapy, histological type of primary tumor, and pathological TNM stage were employed for univariate survival analysis. For FDG PET/CT images, SUVmax, SUVmean, MTV, TLG, and pseudotime were selected as variables. These five parameters were recorded as continuous scales and divided into two groups as a discrete scale according to a cutoff value to best discriminate prognosis of overall survival (OS) in all patients. They were explored by the “surv_cutpoint” function in the package “survminer”. SUVmax, SUVmean, MTV, TLG, and pseudotime were employed with both discrete and continuous scales for univariate survival analysis.

OS and disease-free survival (DFS) were endpoints of analysis. The Cox proportional hazards model was used to evaluate the prognostic power of each variable. Hazard ratios (HRs) and 95% confidence intervals were estimated. Log-rank statistics were also obtained by the Kaplan–Meier method. Significant variables in univariate survival analysis with *p*-values of log-rank statistics lower than 0.05 were included in multivariate survival analysis. Variables with collinearity were excluded. Due to multicollinearity issues, multivariate survival analysis was performed repeatedly according to each image parameter. All statistical analyses were performed using R software (v. 4.0.4, R Foundation for Statistical Computing, Vienna, Austria). A *p*-value lower than 0.05 was considered statistically significant.

## 3. Results

### 3.1. Demographic Data

The clinical characteristics and demographics of the subjects are described in Table 1. Overall, 72.6% of patients were male, and the median age was 62.3 years. The histological type of 63.4% of total subjects was adenocarcinoma. The tumors were located in the right lung in 66.0% of subjects. No adjuvant therapy was performed in 66.0% of subjects. Among clinical stages, 63.0% of subjects were stage IIIA. In post-operative pathological findings, stage III was the most common stage.

### 3.2. Pseudotime Estimation

Pseudotime of the primary tumor was successfully estimated in texture parameter datasets. A principal components analysis was performed to visualize the order of pseudotime in each FDG PET/CT image (Figure 2a) and it demonstrated that pseudotime was estimated according to a specific direction, not randomly. A total of 20 parameters showed positive correlations with pseudotime, and 40 parameters showed negative correlations with pseudotime. The top 10 features displayed in Figure 2b and detailed statistics for all parameters are described in Appendix A. Normalized co-occurrence homogeneity (including representatively), normalized co-occurrence inverse difference moment, black–white symmetry, long run emphasis, and second moment showed positive correlations with pseudotime. Short run emphasis, normalized co-occurrence dissimilarity, short zone emphasis, neighboring gray level dependence entropy, and small number emphasis showed negative correlations with pseudotime. Additionally, we performed correlation analyses between conventional image parameters and pseudotime. SUVmax, SUVmean, MTV, and TLG; all showed positive correlations with pseudotime (Figure 2, r = 0.480, r = 0.401, r = 0.784, r = 0.478, respectively; *p* < 0.001 for all).

### 3.3. Survival Analysis

In univariate survival analysis, sex, age with discrete scale, age with continuous scale, histological type, pT stage, pN stage, pathological TNM stage, SUVmax with discrete scale, SUVmean with discrete scale, and SUVmean with continuous scale were significant prognostic factors for DFS (Table 2). Sex, age with discrete scale, age with continuous scale, pT stage, MTV with discrete scale, MTV with continuous scale, TLG with discrete scale, TLG with continuous scale, pseudotime with discrete scale, and pseudotime with continuous scale were significant prognostic factors for OS (Table 2). In the multivariate survival analysis, pseudotime was selected as an independent prognostic factor for OS (Table 3). In contrast, MTV and TLG were not independent prognostic factors for OS. MTV with discrete scale, TLG with discrete scale, and pseudotime with discrete scale discriminated the risk of overall survival well (Figure 3).

## 4. Discussion

In the present study we found that pseudotime analysis of primary tumors could be successfully applied to radiomics data drawn from FDG PET/CT images in NSCLC. Several texture parameters showed significant correlations with estimated pseudotime. Pseudotime was a significant prognostic factor on both continuous and discrete scales in the univariate survival analysis for OS. In addition, it was an independent prognostic factor in the multivariate analysis for OS, in contrast to MTV and TLG.

Radiomics is a new concept in which image features are extracted from medical images using mathematical algorithms. Various texture parameters can be calculated by radiomics algorithms. Previous studies reported that texture parameters have prognostic significance in lung cancer. Park et al. [7] showed that entropy and homogeneity were significant prognostic factors for progression-free survival of lung cancer patients. Lovinfosse et al. [16] reported that dissimilarity was an independent predictor of outcomes in patients with lung cancer treated with radiotherapy and suggested hypotheses for possible mechanisms explaining how those texture parameters affect the prognosis of lung cancer. However, there are still some limitations to applying radiomics concepts in clinical contexts. Whereas the biological meanings of conventional parameters such as SUVmax or volumetric parameters such as MTV are straightforward, it is difficult to understand the implications of texture parameters as mathematical products. In addition, there is no consensus regarding which texture parameter is most appropriate to utilize and easiest to understand, even among nuclear medicine physicians. Thus, developing an integrated parameter from various texture parameters is necessary.

Pseudotime analysis is a recently emerging method to estimate the genetic or biologic evolution of cells or tissues based on large-scale transcriptomic expression data [8]. This method is based on the hypothesis that data from multiple cross-sectional specimens can be integrated into consecutive datasets reflecting temporal evolution [17]. There have been previous related studies using pseudotime analysis to investigate temporal change of tumor biology (Table 4). Kim et al. [9] attempted to reveal the evolution of malignant cells and immune cells using the Monocle method for single-cell RNA-sequencing data. Pang et al. [18] analyzed the progression of glioblastoma cells using the Monocle method for single-cell RNA-sequencing data. Beyond single-cell RNA-sequencing data, pseudotime analysis has been applied to uncover the temporal evolution of tumor characteristics of colorectal cancer and breast cancer based on tissue-scale RNA-sequencing data using the PhenoPath method [14]. We previously reported the evolution of the clinicomolecular and immunological characteristics of lung adenocarcinoma using pseudotime analysis based on tissue-scale RNA-sequencing data [10]. Transcriptomics data and radiomics data both consist of large-scale datasets with various features from multiple cross-sectional specimens or subjects. However, no previous study has implemented pseudotime analysis in radiomics datasets from FDG PET/CT images. In the present study we applied pseudotime analysis to radiomics data from multiple cross-sectional FDG PET/CT images to estimate the sequential order of individuals for the first time.

There are over 70 methodologies with various characteristics and algorithms to execute pseudotime analysis [19]. Among them, the Monocle and Slingshot methods are widely used methods in single-cell RNA transcriptomic data. Many methods, including these, employ three steps to estimate pseudotime as follows: dimension reduction, clustering, and trajectory estimation. In contrast, PhenoPath, another pseudotime analysis method, uses a Bayesian statistic that integrates linear regression to estimate the ordering of high-dimensional data across individuals [14]. In this study, PhenoPath was selected as an appropriate analytical tool for the following reasons: First, no previous study has applied other popular methods such as Monocle or Slingshot to data other than single-cell RNA-sequencing data. PhenoPath was previously applied for bulk RNA-sequencing data and has proven to be useful to estimate the temporal order of individuals [10,14]. Second, linear regression modeling is required for the application of pseudotime in clinical contexts. Pseudotime is expected to be useful when it provides disease-progression status for patients based on a predictive model. Linear regression is thought to be the simplest and easiest method for constructing a predictive model. Third, a pilot study showed that the Slingshot method cannot estimate pseudotime accurately. Clustering results were too coarse to construct appropriate trajectories (data not shown). Relatively small numbers of patients, relatively small dimensions of radiomics data, and heterogeneous patterns of radiomics data might cause this problem. Therefore, PhenoPath was employed in this study and pseudotime was successfully acquired.

We found that pseudotime showed good associations with SUVmax, SUVmean, MTV, and TLG. It is well known that these parameters increase with cancer stage. Therefore, estimated pseudotime was a significant predictor of disease progression in lung cancer. The results of the present study suggest the following clinical advantages of pseudotime as a parameter: First, pseudotime showed good discrimination of high-risk patients for OS and was an independent prognostic factor for OS, unlike MTV or TLG. This implies that an integrated single parameter derived from multiple texture parameters may be more useful to predict survival prognosis than conventional volumetric parameters. In a previous study, texture parameters showed significant differences between responders and non-responders, in contrast to conventional parameters [20]. Another study revealed that only contrast and coefficient of variation of SUVs had prognostic power for long-term survival in locally advanced lung cancer [21]. According to these studies, we hypothesized that tumor heterogeneity may have closer associations with treatment response and survival than conventional parameters in lung cancer patients with CCRT. Therefore, pseudotime derived from various texture parameters is expected to be useful, especially in locally advanced lung cancer patients. Second, pseudotime based on FDG PET/CT is expected to provide information on disease progression, not only for physicians but also for patients. Explanations with concrete values for disease progression may be useful to communicate with patients and help them to understand their diseases more accurately. Therefore, further study should attempt the prediction of pseudotime in a larger patient pool with machine learning methods. As this study is a preliminary study to demonstrate the possibility and usefulness of pseudotime analysis, predictive analysis was not conducted. More clinical and imaging data are being collected for this further study.

In addition, various image parameters were well correlated with estimated pseudotime. Among texture parameters, it is notable that homogeneity showed positive correlations with pseudotime and MTV. In contrast, dissimilarity showed a negative correlation with pseudotime. There are controversies about associations between the volume and heterogeneity of tumors in FDG PET/CT images. A previous study suggested a negative correlation between tumor volume and uniformity in FDG PET/CT images of lung cancer patients [22]. On the contrary, another study reported that homogeneity and MTV had a significantly positive correlation [16]. The present study echoes the latter finding. Two hypotheses can be suggested. First, tumor characteristics change due to survival of the fittest: tumor cells with the best proliferation and survival become the majority. Second, differences between voxel values of FDG PET/CT can be underestimated due to high cellular density of a voxel or the overall high glucose metabolism of tumor tissue. 

Pseudotime analysis with radiomics data is a newly developed concept in the present study. Therefore, it is important to validate this concept in clinical aspects. First, an external validation is required. A multi-institutional study with more subjects could support the usefulness of pseudotime analysis for radiomics data. Second, a combined modeling for prognosis prediction is needed. Although pseudotime of radiomics data showed good prognostic discrimination in this study, it is not the only factor to determine prognosis. As clinical and pathological factors have been well known to reflect tumor biology, a prognostic model including clinicopathological factors as well as pseudotime from imaging data is expected to have excellent clinical significance. To conduct pseudotime analysis, a general computing system and basic statistical software, such as R, are needed. If there is a large scale of subjects or high-dimension dataset, an advanced computing system may be required. 

This method can be also applied to radiomics data of other image modalities such as magnetic resonance imaging (MRI). For example, an apparent diffusion coefficient (ADC) is a value reflecting cell density based on the motion of water molecules [23]. Radiomics data based on ADC are known to have a correlation with the prognosis of cervical cancer and histological types of head and neck cancer [24,25]. Considering this previous knowledge, pseudotime analysis from an ADC map of an MRI can provide additional information in other cancer subtypes such as cervical cancer, or head and neck cancer. 

This study has several limitations. First, only subjects undergoing neoadjuvant CCRT as an initial treatment were included in this study. As high-stage tumors usually demonstrate high FDG uptake, images of high-stage patients were expected to be useful in this preliminary study due to easy tumor recognition and delineation. Therefore, subjects with neoadjuvant CCRT were recruited for this project. As described earlier, a larger cohort is being collected for further study that will include patients with other treatment options such as surgery only or palliative chemotherapy. Second, it was difficult to evaluate the prognostic power of pseudotime for predicting the complete remission after neoadjuvant CCRT as there were only 29 patients (9.6% of total subjects) with complete remission, and this number was too small to reach statistical significance in this analysis. Finally, PET/CT images were acquired by multiple instruments. As the technical parameters of each instrument are different, harmonization was performed to reduce batch effects. Therefore, the analysis of integrated data is reasonable.

In conclusion, pseudotime analysis was successfully applied to texture parameters from primary tumors on FDG PET/CT images of NSCLC patients for the first time. Pseudotime showed a good stratification power for prognosis and was an independent prognostic factor for OS in NSCLC cancer patients with tri-modality therapy, performing better than conventional PET parameters. We demonstrated the possibility of integrating various texture parameters into a single parameter that reflects disease progression status. Pseudotime, as a concrete measure of disease progression, is expected to be used in clinical contexts to evaluate disease and prognosis.

Our institutional review board (Samsung Medical Center IRB) approved this retrospective cohort study (#2022-01-086), and the requirement for informed consent was waived due to its retrospective design. All data were anonymized.

## Figures and Tables

**Figure 1 cancers-14-03809-f001:**
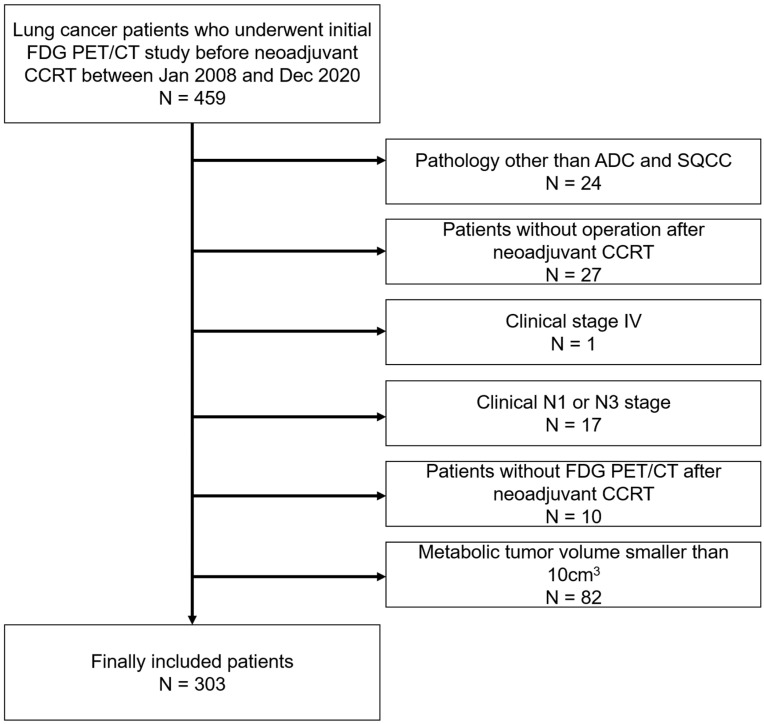
Patient inclusion and exclusion criteria. Four hundred fifty-nine patients were retrospectively enrolled. Among them, patients with pathologies other than adenocarcinoma or squamous cell carcinoma, those who did not undergo curative surgery after neoadjuvant concurrent chemoradiation therapy (CCRT), those with clinical stage IV or clinical stage N1 or N3 disease, those without follow-up FDG PET/CT after neoadjuvant CCRT, and those with tumor volumes smaller than 10 cm^3^ were subsequently excluded. Ultimately, 303 patients were included.

**Figure 2 cancers-14-03809-f002:**
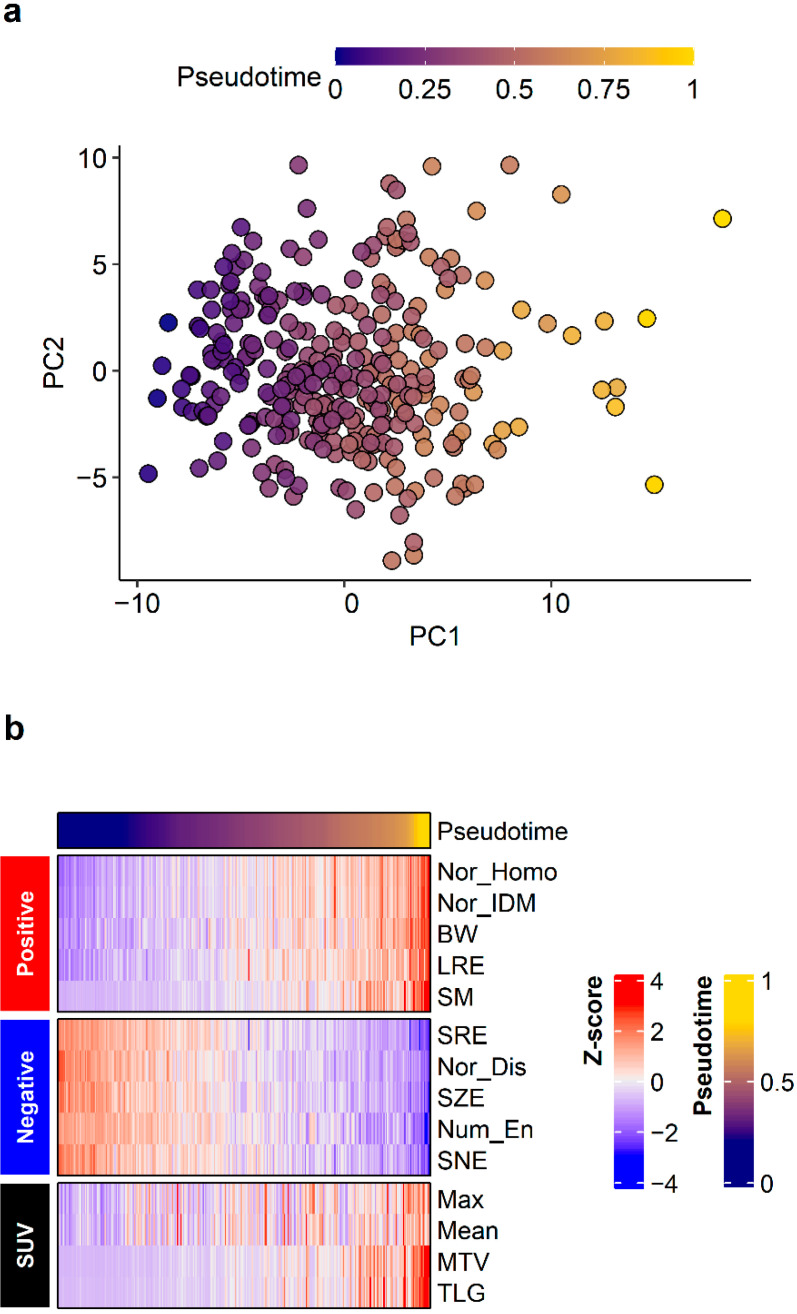
Pseudotime analysis results. Pseudotime of primary tumors was successfully estimated in the radiomics dataset. (**a**) A principal components analysis plot visualized the order of pseudotime in each FDG PET/CT image. Although there was no clustering, pseudotime was estimated according to a specific direction and not randomly. (**b**) The top 10 features demonstrating positive correlations and top 10 features demonstrating negative correlations with pseudotime are shown. Conventional image parameters based on SUV are also shown.

**Figure 3 cancers-14-03809-f003:**
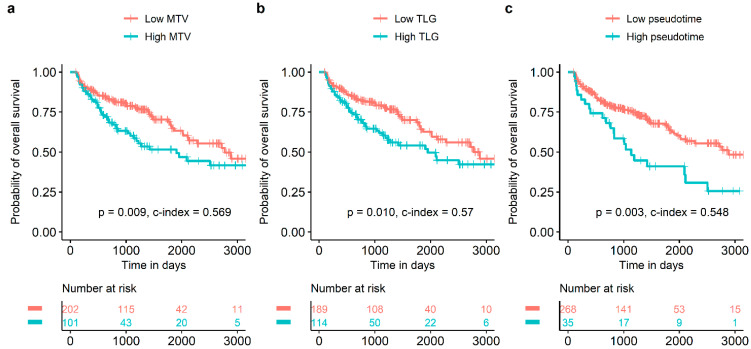
Survival curves according to MTV, TLG, and pseudotime. MTV (**a**), TLG (**b**), and pseudotime (**c**) of the primary tumor were significant prognostic factors for overall survival.

**Table 1 cancers-14-03809-t001:** Demographic and clinical characteristics of patients with lung cancer.

Characteristics	Patients, *n* (%)
Sex	
Female	83 (27.4)
Male	220 (72.6)
Age, median (range), years	62.3 (31.8–79.0)
<59	105 (34.7)
59~66	100 (33.0)
66≤	98 (32.3)
Histological type	
Adenocarcinoma	192 (63.4)
Squamous cell carcinoma	111 (36.6)
Location	
Right lung	200 (66.0)
Left lung	103 (34.0)
Adjuvant therapy	
No	200 (66.0)
Chemotherapy	103 (34.0)
Clinical T stage	
T1	44 (14.5)
T2	148 (48.8)
T3	81 (26.7)
T4	30 (9.9)
Clinical stage	
IIIA	191 (63.0)
IIIB	112 (37.0)
Histological grade	
1	83 (11.5)
2	533 (73.9)
3	105 (14.6)
Unknown	
Post-operative pathological T stage	
0	34 (11.2)
T1	118 (38.9)
T2	103 (34.0)
T3	39 (12.9)
T4	9 (3.0)
Post-operative pathological N stage	
N0	120 (39.6)
N1	24 (7.9)
N2	158 (52.1)
N3	1 (0.3)
Post-operative pathological TNM stage	
0	29 (9.6)
I	64 (21.1)
II	43 (14.2)
III	165 (54.5)
IV	2 (0.7)
SUVmax, median (range)	13.2 (4.4–32.8)
<14.7	194 (64.0)
14.7≤	109 (36.0)
SUVmean, median (range)	4.9 (2.6–11.5)
<3.9	62 (20.5)
3.9≤	241 (79.5)
MTV, median (range), cm3	40.5 (10.1–468.8)
<60.2	202 (66.7)
60.2≤	101 (33.3)
TLG, median (range)	192.7 (28.7–2554.4)
<272.4	189 (62.4)
272.4≤	114 (37.6)
Pseudotime, median (range)	0.38 (0–1)
<0.38	268 (88.4)
0.38≤	35 (11.6)
Instrument	
Discovery LS	67 (22.1)
Discovery STE	216 (71.3)
Discovery MI DR	20 (6.6)

TNM: tumor-node-metastasis; SUVmax: maximum standardized uptake value; SUVmean: mean standardized uptake value; MTV: metabolic tumor volume; TLG: total lesion glycolysis.

**Table 2 cancers-14-03809-t002:** Univariate Cox regression analysis for survival.

Variable	Categories	Disease-Free Survival				Overall Survival			
		Hazard Ratio	95% Confidence Interval	*p*	*p* of Log-Rank Test	Hazard Ratio	95% Confidence Interval	*p*	*p* of Log-Rank Test
Sex	Male vs. female	1.519	1.084–2.128	0.015	0.01	0.549	0.343–0.878	0.012	0.01
Age	<59				0.01				0.001
	59~66	0.667	0.458–0.972	0.035		0.987	0.613–1.675	0.960	
	66≤	0.556	0.366–0.843	0.006		2.023	1.284–3.186	0.002	
Age (1-yr increase)		0.976	0.959–0.993	0.007	0.007	1.036	1.012–1.061	0.003	0.003
Location	Right				0.7				0.9
	Left	1.080	0.769–1.516	0.659		1.018	0.683–1.517	0.930	
Histological type	Adenocarcinoma				<0.001				0.8
	Squamous cell carcinoma	0.379	0.254–0.567	<0.001		1.044	0.702–1.551	0.833	
Clinical T stage	T1				0.07				0.4
	T2	0.817	0.517–1.290	0.386		0.874	0.495–1.545	0.644	
	T3	1.053	0.642–1.728	0.839		1.203	0.659–2.194	0.548	
	T4	0.408	0.184–0.905	0.027		1.400	0.672–2.918	0.369	
Clinical stage	IIIA				1				0.07
	IIIB	0.990	0.704–1.392	0.952		1.415	0.964–2.077	0.076	
Adjuvant therapy	No				0.6				0.2
	Chemotherapy	1.106	0.792–1.545	0.555		1.330	0.899–1.967	0.153	
Histological grade	Well differentiated				0.8				0.8
	Moderately differentiated	1.762	0.245–12.670	0.573		0.654	0.159–2.693	0.556	
	Poorly differentiated	1.786	0.247–12.910	0.565		0.673	0.162–2.801	0.586	
Post-operative pathological T stage	0				<0.001				0.005
	T1	3.470	1.591–7.567	0.002		1.653	0.739–3.698	0.221	
	T2	2.616	1.185–5.777	0.017		1.629	0.725–3.663	0.238	
	T3	2.576	1.059–6.264	0.037		2.347	0.980–5.623	0.056	
	T4	7.714	2.702–22.022	<0.001		5.696	1.993–16.282	0.001	
Post-operative pathological N stage	N0				<0.001				0.7
	N1	1.500	0.785–2.865	0.219		1.151	0.587–2.257	0.682	
	N2	2.200	1.512–3.200	<0.001		1.138	0.754–1.715	0.538	
	N3	2.293	0.315–16.715	0.413		2.957	0.405–21.618	0.285	
Post-operative pathological stage	0				<0.001				0.6
	I	2.398	0.915–6.286	0.075		1.694	0.683–4.203	0.255	
	II	2.840	1.054–7.653	0.039		2.117	0.840–5.337	0.112	
	III	4.598	1.871–11.304	0.001		1.931	0.834–4.472	0.125	
	IV	2.168	0.253–18.563	0.480		2.400	0.288–20.015	0.418	
SUVmax	<14.7				0.4				0.2
	14.7≤	0.858	0.607–1.213	0.386		1.260	0.855–1.856	0.243	
SUVmax (continuous)		0.959	0.924–0.994	0.024	0.02	1.013	0.973–1.055	0.535	0.5
SUVmean	<3.9				0.03				
	3.9≤	0.662	0.454–0.965	0.032		0.678	0.440–1.044	0.077	0.08
SUVmean (continuous)		0.854	0.748–0.976	0.020	0.02	0.986	0.850–1.144	0.855	0.9
MTV	<60.2				0.9				
	60.2≤	0.985	0.693–1.400	0.931		1.663	1.133–2.439	0.009	0.009
MTV (continuous)		0.997	0.994–1.000	0.093	0.09	1.004	1.001–1.007	0.004	0.003
TLG	<272.4				0.8				
	272.4 ≤	0.953	0.676–1.342	0.782		1.619	1.107–2.368	0.013	0.01
TLG (continuous)		1.000	0.999–1.000	0.094	0.09	1.001	1.000–1.001	0.016	0.01
Pseudotime	<0.59				0.4				0.003
	0.59≤	1.196	0.787–1.816	0.402		1.894	1.236–2.901	0.003	
Pseudotime (continuous)		0.694	0.277–1.739	0.436	0.4	3.085	1.108–8.588	0.031	0.03

SUVmax: maximum standardized uptake value; SUVmean: mean standardized uptake value; MTV: metabolic tumor volume; TLG: total lesion glycolysis.

**Table 3 cancers-14-03809-t003:** Multivariate Cox regression analysis for overall survival.

		MTV			TLG			Pseudotime		
Variable	Categories	Hazard Ratio	95% Confidence Interval	*p*	Hazard Ratio	95% Confidence Interval	*p*	Hazard Ratio	95% Confidence Interval	*p*
Sex	Female vs. male	0.626	0.387–1.011	0.056	0.611	0.380–0.985	0.043	0.605	0.376–0.974	0.038
Age	<59									
	59~66	0.929	0.560–1.541	0.774	0.952	0.575–1.577	0.848	1.014	0.612–1.680	0.957
	66≤	1.775	1.118–2.819	0.015	1.807	1.141–2.863	0.012	2.060	1.298–3.269	0.002
MTV	<60.2									
	60.2≤	1.448	0.975–2.149	0.066						
TLG	<272.4									
	272.4≤				1.459	0.992–2.144	0.055			
Pseudotime	<0.59									
	0.59≤							2.245	1.397–3.609	<0.001

MTV: metabolic tumor volume; TLG: total lesion glycolysis.

**Table 4 cancers-14-03809-t004:** A benchmarking table of previous studies.

Study	Subject Cancer Type	Subject Data Type	Analysis Method
Kim et al. [9]	Lung cancer (adenocarcinoma)	Single-cell RNA sequencing	Monocle
Pang et al. [18]	Glioblastoma	Single-cell RNA sequencing	Monocle
Campbell and Yau [14]	Colorectal cancer and breast cancer	Tissue-scale RNA sequencing	Phenopath
Lee et al. [10]	Lung cancer (adenocarcinoma)	Tissue-scale RNA sequencing	Phenopath
The present study	Lung cancer (both adenocarcinoma and squamous cell carcinoma)	Radiomics data from FDG PET/CT images	Phenopath

## Data Availability

The data used in this study are available from the corresponding author upon reasonable request.

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
