# Peer review of "Prognostic Significance of Pseudotime from Texture Parameters of FDG PET/CT in Locally Advanced Non-Small-Cell Lung Cancer with Tri-Modality Therapy"

_cancers, 2022, doi:10.3390/cancers14153809_

Round 1
Reviewer 1 Report
The authors successfully applied the pseudotime analysis to the texture parameters from NSCLC tumors FDG-PET/CT images, which reflect the significant development of this field. The manuscript is well written and organized. We recommend for publication in present form.
Author Response
The authors successfully applied the pseudotime analysis to the texture parameters from NSCLC tumors FDG-PET/CT images, which reflect the significant development of this field. The manuscript is well written and organized. We recommend for publication in present form.
Thank you for your comment.
Reviewer 2 Report
The authors proposed the solutions for overall survival in locally advanced non-small cell lung cancer (NSCLC) patients as a prognostic measure of pseudotime from texture parameters of FDG PET/CT with tri-modality therapy. However, the author needs to focus on some of the points identified as follows.
1. Comparison with other similar work: The benchmarking table representing the comparisons with other similar work is missing.
2. Related Work: The authors do not consider the related work as a reference for the proposed work.
3. Clinical validation:
The author should discuss the clinical validation of the correctness of the proposed system.
4. Implementation Requirement: The hardware and software implementation requirements of the proposed system should be provided.
5. Discussion, Future Work, and Extension:
The discussion, future scope, and extension of the current work should be more elaborative. What should be the effect of other features apart from texture features on pseudotime analysis?. What should be the effect, if other image modalities are considered apart from FDG PET/CT. What should be the impact of the proposed solutions on other types of cancers other than non-small cell lung cancer (NSCLC)?. Therefore, the discussion should consider all the multiple parameters in the prospect.
In the discussion section, citation 7 should be just after Park et al., not at the end of the statement. Same for Lovinfosse et al.
Author Response
The authors proposed the solutions for overall survival in locally advanced non-small cell lung cancer (NSCLC) patients as a prognostic measure of pseudotime from texture parameters of FDG PET/CT with tri-modality therapy. However, the author needs to focus on some of the points identified as follows.
- Comparison with other similar work:
The benchmarking table representing the comparisons with other similar work is missing.
- Related Work:
The authors do not consider the related work as a reference for the proposed work.
Another related work was added in Discussion section paragraph 3 with red color texts. An explanation for other similar works and a benchmarking table were also added in Discussion section paragraph 3 and Table 4 as your comment.
- Clinical validation:
The author should discuss the clinical validation of the correctness of the proposed system.
- Implementation Requirement:
The hardware and software implementation requirements of the proposed system should be provided.
An additional discussion for the clinical validation strategy and system requirements of the proposed system were described in Discussion section paragraph 7.
- Discussion, Future Work, and Extension:
The discussion, future scope, and extension of the current work should be more elaborative. What should be the effect of other features apart from texture features on pseudotime analysis?. What should be the effect, if other image modalities are considered apart from FDG PET/CT. What should be the impact of the proposed solutions on other types of cancers other than non-small cell lung cancer (NSCLC)?. Therefore, the discussion should consider all the multiple parameters in the prospect.
Another feature such as ADC can be also analyzed with pseudotime method. It is derived from MRI, image modality other than FDG PET/CT. An application of pseudotime analysis in MRI can provide pathological or prognostic information for other diseases such as cervical cancer or head and neck cancer, cancer subtypes that MRI is commonly used for initial staging. This point was additionally described in Discussion section paragraph 8.
In the discussion section, citation 7 should be just after Park et al., not at the end of the statement. Same for Lovinfosse et al.
The order was adjusted as your comment.
Thank you for your helpful comments.
Reviewer 3 Report
This is a well written manuscript. I only have minor comments:
1. Since the methodology is relatively new in applying the pseudotime analysis to radiomics features, the authors should describe the analysis methodology in more details.
2. From figure 3, I could not see why the pseudotime performed better than the MTV and TLG. Also, I could not see why the MTV and TLG are multi-variate variables.
Author Response
This is a well written manuscript. I only have minor comments:
- Since the methodology is relatively new in applying the pseudotime analysis to radiomics features, the authors should describe the analysis methodology in more details.
We added some more description of analysis methods. Also, statistical knowledge related to this methodology was additionally described in Methods section. The revised points in the manuscript are in red color in the revised manuscript.
- From figure 3, I could not see why the pseudotime performed better than the MTV and TLG. Also, I could not see why the MTV and TLG are multi-variate variables.
In Figure 3, all of MTV, TLG, and pseudotime showed prognostic discrimination. As you can see in Table 3, we made three multivariate Cox regression models including each of MTV, TLG, and psuedotime respectively due to multicollinearity issue. In these analyses, pseudotime was selected as an independent prognostic factor in contrast to MTV or TLG. Considering both of survival analysis and multivariate analysis, we suggested the prognostic power of pseudotime was better than conventional parameters. Considering only figure 3, there is no definite evidence to suggest superiority of pseudotime as your comment.
MTV and TLG was selected as significant prognostic factors for overall survival in univariate analysis. Therefore, those parameters were included in multivariate analysis of each prognostic model.
Thank you for your helpful comments.